# Effect of Exercise Habit on Skeletal Muscle Mass Varies with Protein Intake in Elderly Patients with Type 2 Diabetes: A Retrospective Cohort Study

**DOI:** 10.3390/nu12103220

**Published:** 2020-10-21

**Authors:** Yoshitaka Hashimoto, Ayumi Kaji, Ryosuke Sakai, Fuyuko Takahashi, Rena Kawano, Masahide Hamaguchi, Michiaki Fukui

**Affiliations:** Department of Endocrinology and Metabolism, Graduate School of Medical Science, Kyoto Prefectural University of Medicine, 465 Kajii-cho, Kawaramachi-Hirokoji, Kamigyo-ku, Kyoto 602-8566, Japan; kaji-a@koto.kpu-m.ac.jp (A.K.); sakaryo@koto.kpu-m.ac.jp (R.S.); fuyuko-t@koto.kpu-m.ac.jp (F.T.); rena0421@koto.kpu-m.ac.jp (R.K.); mhama@koto.kpu-m.ac.jp (M.H.); michiaki@koto.kpu-m.ac.jp (M.F.)

**Keywords:** exercise, protein, muscle mass, type 2 diabetes mellitus, sarcopenia, elderly

## Abstract

Exercise has been reported to be effective in maintaining and recovering muscle; however, the effect of exercise combined with adequate or inadequate protein intake on muscle mass is not clear. Therefore, this study investigates the effect of exercise habit on changes in muscle mass, with adequate or inadequate protein intake. This retrospective cohort study included 214 elderly patients with type 2 diabetes. The rate of skeletal muscle mass index (SMI) change (%) was defined as ((SMI at follow-up minus SMI at baseline)/(follow-up years [kg/m^2^/year] × SMI at baseline [kg/m^2^])) × 100. Adequate protein intake was defined as protein intake ≥1.2 g/kg ideal body weight/day. During a mean follow-up duration of 18.0 (7.1) months, the rate of SMI change was −1.14 (4.10)% in the whole sample. The rate of SMI change of non-habitual exercisers with inadequate protein intake, habitual exercisers with inadequate protein intake, non-habitual exercisers with adequate protein intake, and habitual exercisers with adequate protein intake was −1.22 (3.71), −2.31 (3.30), −1.88 (4.62), and 0.36 (4.29)%, respectively. Compared with patients with exercise habit and adequate protein intake, the odds ratio for decreasing SMI was 2.50 (0.90–6.90, *p* = 0.078) in patients with no exercise habit and inadequate protein intake, 3.58 (1.24–10.4, *p* = 0.019) in those with exercise habit and inadequate protein intake, and 3.03 (1.27–7.22, *p* = 0.012) in those with no exercise habit and adequate protein intake, after adjusting for covariates. Therefore, exercise habit without adequate protein intake was associated with an increased risk of decreasing SMI compared with exercise habit with adequate protein intake.

## 1. Introduction

The number of elderly patients with type 2 diabetes mellitus (T2DM) is increasing, often accompanied by sarcopenia [1,2,3]. Sarcopenia, defined as the age-associated loss of muscle mass, strength, and function [1,2], is now known to be a risk factor of cardiovascular disease and mortality [4,5,6]. In addition, diabetes accelerates the reduction of muscle mass and strength due to inflammatory cytokines, insulin resistance, hyperglycemia, and endocrine changes [1,2]. Reduced insulin signaling leads to decreased protein synthesis and increased protein degradation [1,2], and insulin resistance leads to muscle wasting [7], which leads to muscle mass reduction. Therefore, prevention of muscle loss is an important treatment for elderly patients with T2DM.

To prevent sarcopenia, both adequate protein intake [8,9] and exercise are recommended [10,11]. For adults over 65 years of age, a dietary protein intake of 1.0 to 1.2 g/kg ideal body weight (IBW)/day is recommended to maintain and recover muscle [12,13]. In fact, insufficient dietary protein intake has been associated with a loss of muscle mass [13]. In addition, exercise has been reported to be effective in maintaining and recovering muscle [14,15].

However, the effect of exercise on muscle mass considering different amounts of protein intake is unknown. Therefore, this retrospective cohort study of elderly patients with type 2 diabetes investigates the effect of exercise habit on changes in skeletal muscle mass, with and without adequate protein intake.

## 2. Materials and Methods

### 2.1. Study Patients

The KAMOGAWA-DM cohort study, initiated in 2014, is an ongoing cohort study to clarify the natural history of diabetes [16]. This cohort study includes outpatients at the Department of Endocrinology and Metabolism at Kyoto Prefectural University of Medicine (KPUM) Hospital, and the Department of Diabetology at Kameoka Municipal Hospital. All patients gave written informed consent. This study was approved by the KPUM Ethics Committee (No. RBMR-E-466-5) and carried out accordance with the Declaration of Helsinki. The inclusion criteria of this study were follows: patients aged 65 years and older, with T2DM, who completed the Brief-Type Self-administered Diet History Questionnaire (BDHQ) from January 2016 to April 2018 [17]. The exclusion criteria were incomplete questionnaires, extremely low or high energy intake (<600 or >4000 kcal/day) [18], unreliable data, and lack of bioimpedance analysis (BIA) at baseline examination. Patients who had no follow-up BIA data more than 6 months after the baseline examination were also excluded.

### 2.2. Lifestyle, Medications, and Laboratory Data Collection

The detailed questionnaire about exercise habit included the following question, “Do you perform exercise, including recreational and sports activities such as walking, jogging, golf, ball game, soccer, cycling, swimming, combat sports, gym training, and so on? If yes, please indicate the types of recreational and sports activities and their frequency and time.” In this study, we defined the respondent as having an exercise habit if they performed any of these physical activities at least once a week. This definition was validated previously [19]. Moreover, participants were classified as non-smokers or current smokers. Data on medication, including insulin, were obtained from medical records. Blood samples were collected in the morning for biochemical measurements. Hemoglobin A1c (HbA1c) was evaluated by high performance liquid chromatography. Estimated glomerular filtration rate (eGFR) was calculated with the equation of the Japanese Society of Nephrology: eGFR = 194 × Cre^−1.094^ × age^−0.287^ (mL/min/1.73 m^2^) (for women, ×0.739) [20].

### 2.3. Measurement of Body Composition Determined by Bioelectric Impedance

The participants’ body composition was evaluated with a multifrequency impedance BIA, InBody 720 (InBody Japan, Tokyo, Japan) [21], reported as having a good correlation with dual-energy X-ray absorptiometry (DEX). The R^2^ coefficients of BIA and DEX were 0.88 for men and 0.83 for women [20]. Data for body weight (BW, kg), appendicular muscle mass (kg), and fat mass (kg) were obtained using this analyzer. Body mass index (BMI, kg/m^2^) and skeletal muscle mass index (SMI, kg/m^2^) were calculated as body weight (kg) divided by height squared (m^2^), and as appendicular muscle mass (kg) divided by height squared (m^2^), respectively [22]. Percent fat mass (%) was defined as (fat mass (kg) ÷ body weight (kg)) × 100. In this study, change in SMI (kg/m^2^/month) was calculated as (follow-up SMI [kg/m^2^] − baseline SMI [kg/m^2^]) ÷ follow-up period (month). Next, the rate of SMI change (%) was calculated as the change in SMI (kg/m^2^/month) × (12 ÷ baseline examination SMI [kg/m^2^]) × 100. In addition, SMI decrease was defined as a SMI change rate of less than 0.5%. This is reported as average rate of muscle loss [23]. Ideal body weight (IBW) was calculated as 22 × patient height squared (m^2^) [24].

### 2.4. Estimation and Assessment of Habitual Food and Nutrient Intake

Habitual food and nutrient intake was evaluated with the BDHQ [18], which estimates the dietary intake and variations over the past month of 58 food items. The BDHQ consists of five sections: (i) intake frequency of 46 foods and non-alcoholic items; (ii) daily intake of rice, including type of rice and miso soup; (iii) frequency and quantity of alcohol consumption, from a list of five types of alcoholic beverages; (iv) usual cooking methods; and (v) general dietary behavior. Estimates of 58 food item intake, and energy, carbohydrate, protein, and fat intake were calculated with an algorithm based on the Standard Tables of Food Composition in Japan [25]. The median Spearman’s correlation coefficients for the BDHQ and semi-weighed dietary records were 0.48 for men and 0.44 for women [26]. These BDHQ data was used to estimate the intake of total dietary energy (kcal/day), carbohydrates (g/day), total protein (g/day), animal protein intake (g/day), vegetable protein intake (g/day), total fat (g/day), and ethanol (g/day). Energy intake (kcal/kg IBW/day) was calculated by dividing total dietary energy (kcal/day) by IBW (kg). Protein intake (g/kg IBW/day) was calculated by dividing total protein intake (g/day) by IBW (kg) [27]. Adequate protein intake was defined as protein intake ≥1.2 g/kg IBW/day [8]. Habitual alcohol intake was defined as ethanol intake ≥30 g/day for men, and ≥20 g/day for women [28].

### 2.5. Statistical Analysis

Categorical and continuous variables were represented as counts and mean (standard deviation), respectively. The difference between groups was evaluated using Student’s *t*-test, one-way analysis of variance (ANOVA) and post hoc Tukey–Kramer test, or chi-square test. We investigated the association between animal protein, vegetable protein, carbohydrate, and fat intake and rate of SMI change using Pearson’s correlation coefficient. Furthermore, we also investigated these associations according to the presence or absence of exercise habit using Pearson’s correlation coefficient. The combined effect of presence or absence of exercise habit and/or adequate protein intake on decreasing SMI was evaluated by logistic regression, adjusting for gender, age, smoking status, habitual alcohol consumption, duration of diabetes, SMI at baseline, BMI, energy intake, animal protein intake, vegetable protein intake, carbohydrate intake, and fat intake. Next, a multiple regression analysis was performed to evaluate the effect of exercise habit on the rate of SMI change, adjusting for gender, age, smoking status, habitual alcohol consumption, duration of diabetes, SMI at baseline BMI, energy intake, animal protein intake, vegetable protein intake, carbohydrate intake, and fat intake, with and without an adequate protein intake of ≥1.2 g/kg IBW/day [8].

## 3. Results

This retrospective cohort study initially included 261 patients; however, 53 patients were excluded. Thus, the final sample consisted of 214 participants (Figure 1).

Participants’ baseline characteristics are shown in Table 1. Mean age, BMI, and SMI were 72.2 (5.1) years, 23.7 (3.9) kg/m^2^, and 6.8 (0.9) kg/m^2^, respectively. During the mean follow-up period of 18.0 months (SD 7.1), SMI decreased by 0.007 kg/m^2^/month (SD 0.023), and the rate of SMI change was −1.14% (SD 4.10). The nutritional intake of patients with adequate protein was higher than those of patients without.

Figure 2 shows the difference in the rate of SMI change (%) between patients with and without exercise habit and adequate protein intake. There was no difference in the rate of SMI change between patients with and without exercise habit (−0.87% (SD 4.08) vs. −1.51% (SD 4.12), *p* = 0.255) (Figure 2A). On the other hand, there was a significant difference in the rate of SMI change between patients with and without exercise habit and adequate protein intake (*p* = 0.002). Interestingly, the rate of SMI change in habitual exercisers with inadequate protein intake (−2.26% (SD 3.59), *p* = 0.017) and non-habitual exercisers with adequate protein intake (−1.88% (SD 4.62), *p* = 0.028) was significantly worse than that in habitual exercisers with adequate protein intake (0.36% (SD 4.29)) (Figure 2B).

In addition, there was an association of animal protein (kcal/IBW/day) (r = 0.19, *p* = 0.043), vegetable protein (kcal/IBW/day) (r = 0.14, *p* = 0.039), carbohydrate (kcal/IBW/day) (r = 0.12, *p* = 0.069), and fat intake (kcal/IBW/day) (r = 0.21, *p* = 0.002) with rate of SMI change. Furthermore, we also investigated these associations according to the presence or absence of exercise habit (Figure 3). There was no association between nutrient intakes and rate of SMI change among the participants without exercise habit. On the other hand, there was an association between nutrient intake and rate of SMI change among the participants with exercise habit.

Table 2 shows the combined effect of presence/absence of exercise habit and/or adequate protein intake on decreasing SMI. Compared with patients with exercise habit and adequate protein intake, the odds ratio for decreasing SMI was 2.50 (0.90–6.90, *p* = 0.078) in patients with no exercise habit and inadequate protein intake, 3.58 (1.24–10.4, *p* = 0.019) in patients with exercise habit and inadequate protein intake, and 3.03 (1.27–7.22, *p* = 0.012) in patients with no exercise habit and adequate protein intake, after adjusting for covariates.

Finally, to investigate the effect of exercise habits on the rate of SMI change, a multiple regression analysis was performed according to protein intake, as shown in Table 3. Among participants with adequate protein intake, exercise habit was positively correlated with the rate of SMI change (standardized β = 0.255, *p* = 0.011). On the other hand, among the participants without adequate protein intake, exercise habit was negatively correlated with the rate of SMI change (standardized β = −0.182, *p* = 0.094), although this did not reach statistical significance. In addition, although it did not reach statistical significance, energy intake tended to be positively correlated with the rate of SMI change, and carbohydrate intake tended to be negatively correlated with the rate of SMI change.

## 4. Discussion

This retrospective cohort study investigated whether the effect of exercise habit on SMI change rate varied with protein intake. Our results revealed that exercise habit without adequate protein intake was associated with a higher risk of decreasing SMI compared with exercise habit with adequate protein intake. Interestingly, the effect of exercise habit on the rate of SMI change was reversed according to protein intake, and exercise habit was associated with accelerated muscle mass reduction in patients without adequate protein intake.

Muscle mass decreases with age, especially in elderly persons, by 0.5% to 1.0% per year [23]. This study found a 1.14% SMI decrease rate in the whole sample, similar to a previous study [23]. On the other hand, the SMI decrease rate was over 2% in patients with exercise habit and inadequate protein intake and 1.88% in patients with no exercise habit and adequate protein intake, which were higher than average. Therefore, it should be noted that these patients have an increased risk of muscle decline. To prevent this, both adequate protein intake [8,9,12,13] and exercise are recommended [10,11,14]. Without considering protein intake, exercise habit was not associated with the prevention of muscle decline. However, considering both protein intake and exercise, patients with both adequate protein intake and an exercise habit may show lower muscle decline. Interestingly, patients who had an exercise habit but inadequate protein intake had reduced muscle mass.

A possible explanation for the reversal effect of exercise habit on the rate of SMI change by protein intake is as follows. A balance between muscle protein synthesis (MPS) and muscle protein breakdown (MPB) is important for maintaining muscle mass. Proteins are involved in the activation of muscle protein synthesis [29,30]. Further, elderly persons may need more amino acids than younger generations to stimulate MPS [29]. Both resistance and non-resistance training exercise raise both MPS and MPB after exercise, without protein supplementation [31]. Therefore, exercise and inadequate protein intake lead to muscle mass depletion. On the other hand, when supplied with amino acids or proteins, the rate of MPS exceeds that of MPB [32], allowing the maintenance of muscle mass. In summary, maintaining muscle mass requires both exercise habit and adequate protein intake. However, further studies are needed to obtain more advanced knowledge regarding the association between protein intake, exercise, and muscle mass.

In addition, although it did not reach statistical significance, energy intake tended to be positively correlated with the rate of SMI change. Total energy intake was associated with muscle mass. Energy restriction leads to reduced muscle mass size by altering the Akt dependent signaling pathway [33] and to enhanced fibrosis of muscle mass by upregulating the p38 signaling pathway [34]. Moreover, reduced energy intake causes a reduction in protein synthesis. In fact, we previously reported that reduced energy intake was associated with the prevalence of sarcopenia [2]. In addition, there was no association between nutrient intake and rate of SMI change among the participants with no exercise habit. On the other hand, there was an association between nutrient intake and rate of SMI change among the participants with exercise habit. These results indicated that nutrient intakes, including protein intake, were positively associated with rate of SMI change among the habitual exercisers, whereas nutrient intakes were not associated with rate of SMI change among the non-habitual exercisers. In other words, adequate nutrient intakes, including protein intake, are needed to increase SMI, especially in the presence exercise habit.

This study has some limitations. First, this was a retrospective cohort study. Therefore, there may be unknown confounders. In addition, mean study duration was 18 months, and this follow-up duration might not have been enough. Second, there is inaccuracy associated with dietary questionnaires involving respondents’ recall, although the BDHQ showed reasonable validity with semi-weighed dietary records [26]. Third, the lack of detailed information about the frequency, intensity, duration, and type of exercise performed by the respondents is also a limitation of this study, as the frequency, intensity, duration, and type of exercise has profoundly different effects on the preservation and increase of muscle mass. Fourth, the timing and quality of protein intake, which are also an important factor for sarcopenia [35], was not evaluated. However, we showed that the presence or absence of adequate daily protein intake had an effect on changes in muscle mass. Lastly, the number of participants was not large; thus, further studies with larger numbers of participants are needed.

## 5. Conclusions

In conclusion, this study shows that exercise habit without adequate protein intake was associated with a higher risk of decreasing SMI compared with exercise habit with adequate protein intake. Therefore, elderly patients with an exercise habit without adequate protein intake should be aware of the risk of muscle mass loss.

## Figures and Tables

**Figure 1 nutrients-12-03220-f001:**
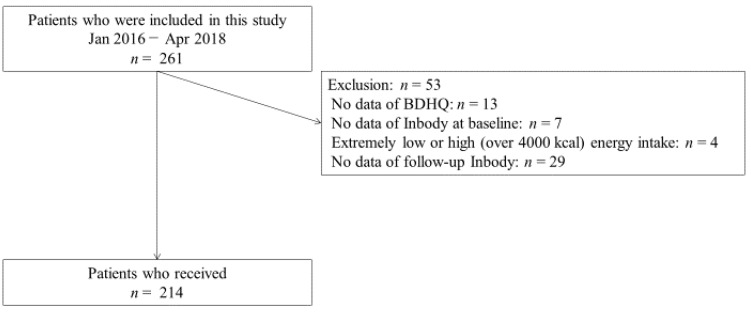
Inclusion and exclusion flow of study participants. BDHQ, Brief-Type Self-administered Diet History Questionnaire.

**Figure 2 nutrients-12-03220-f002:**
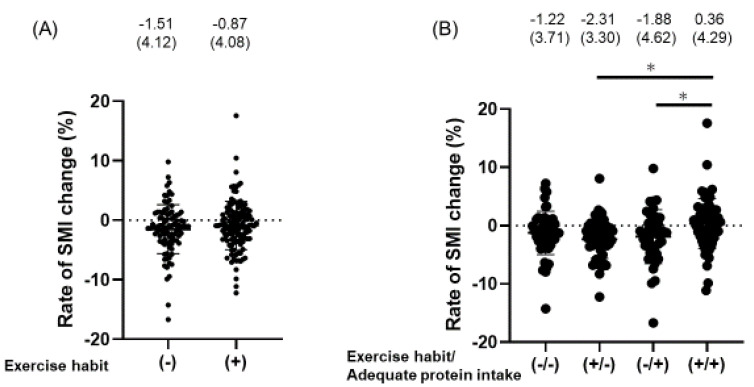
The difference of the rate of SMI change (%). Rate of skeletal muscle mass index (SMI) change (%) was calculated as ((SMI at follow-up minus SMI at baseline)/(follow-up years [kg/m^2^/year] × SMI at baseline [kg/m^2^])) × 100. (**A**) Difference in the rate of SMI change (%) between patients with exercise habit and those without. The difference was evaluated using Student’s t-test. (**B**) Difference in the rate of SMI change (%) between patients with/without exercise habit and adequate protein intake (protein intake ≥1.2 g/kg ideal body weight/day). The difference was evaluated with one-way ANOVA (*p* = 0.018) and Tukey-Kramer test. * *p* < 0.05.

**Figure 3 nutrients-12-03220-f003:**
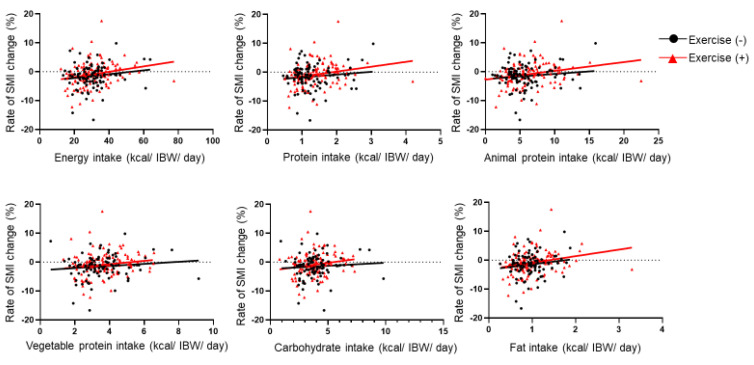
The association between rate of SMI change and nutrient intake according to the exercise habit. SMI, skeletal muscle mass index; and IBW, ideal body weight. Rate of SMI change (%) was calculated as ((SMI at follow-up minus SMI at baseline)/(follow-up years [kg/m^2^/year] × SMI at baseline [kg/m^2^])) × 100. Association between rate of SMI change and diet intakes according to exercise habit using Pearson’s correlation coefficient. Black line represents correlation coefficient for no exercise habit, and red line represents correlation coefficient for exercise habit. The association between rate of SMI change and total energy (kcal/IBW/day), protein (kcal/IBW/day), animal protein (kcal/IBW/day), vegetable protein (kcal/IBW/day), carbohydrate (kcal/IBW/day), and fat intake (kcal/IBW/day) among the participants without exercise habit was given by r = 0.15 (*p* = 0.150), r = 0.12 (*p* = 0.253), r = 0.12 (*p* = 0.268), r = 0.07 (*p* = 0.489), and r = 0.14 (*p* = 0.169), respectively. On the other hand, the association between rate of SMI change and total energy (kcal/IBW/day), protein (kcal/IBW/day), animal protein (kcal/IBW/day), vegetable protein (kcal/IBW/day), carbohydrate (kcal/IBW/day), and fat intake (kcal/IBW/day) among the participants with exercise habit was given by r = 0.23 (*p* = 0.010), r = 0.24 (*p* = 0.008), r = 0.16 (*p* = 0.086), r = 0.17 (*p* = 0.061), and r = 0.24 (*p* = 0.008), respectively.

**Table 1 nutrients-12-03220-t001:** Clinical characteristics of study participants with and without exercise habit and adequate protein intake.

	All *n* = 214	Exercise (−)/Adequate Protein Intake (−) *n* = 52	Exercise (+)/Adequate Protein Intake (−) *n* = 56	Exercise (−)/Adequate Protein Intake (+) *n* = 40	Exercise (+)/Adequate Protein Intake (+) *n* = 66	*p*
Men/Women	120 94	27/25	34/22	21/19	38/28	0.772
Age, years	72.2 (5.1)	72.9 (5.7)	71.5 (4.9)	72.3 (4.6)	72.2 (5.2)	0.612
Duration of diabetes, years	15.6 (10.2)	17.2 (11.5)	15.1 (9.5)	14.5 (11.5)	15.6 (8.7)	0.602
Height, cm	159.9 (8.7)	159.7 (8.1)	161.6 (8.5)	158.5 (9.9)	159.4 (8.5)	0.320
Body weight, kg	60.5 (10.8)	61.5 (10.7)	60.6 (11.0)	59.6 (10.7)	60.1 (11.0)	0.857
Body mass index, kg/m^2^	23.7 (3.9)	24.1 (4.1)	23.2 (4.1)	23.8 (4.2)	23.6 (3.3)	0.662
Insulin (−/+)	154/60	36/16	40/16	27/13	51/15	0.678
Antihypertension medication (−/+)	84/130	19/33	20/36	15/25	30/36	0.666
Antilipidemic medication (−/+)	102/112	21/31	26/30	23/17	32/34	0.439
Smoking (−/+)	184/30	44/8	52/4	34/6	54/12	0.354
Habitual alcohol intake (−/+)	195/19	49/3	51/5	34/6	61/5	0.455
Habit of exercise (−/+)	92/112	52/0	0/56	40/0	0/66	<0.001
Hemoglobin A1c, %	7.2 (1.0)	7.0 (0.9)	7.1 (0.8)	7.3 (1.2)	7.2 (1.0)	0.642
Hemoglobin A1c, mmol/mol	54.8 (10.7)	53.4 (10.0)	54.5 (9.1)	56.0 (13.1)	55.4 (10.8)	0.642
Plasma glucose, mmol/l	8.1 (2.8)	8.0(3.1)	8.1 (2.6)	8.6 (3.8)	7.9 (2.0)	0.627
Creatinine, umol/l	73.8 (25.5)	77.2 (22.5)	74.5 (26.2)	72.2 (24.9)	71.7 (27.8)	0.672
eGFR, ml/min/1.73 m^2^	66.4 (17.8)	61.4 (17.1)	66.8 (17.2)	67.2 (18.7)	69.5 (17.7)	0.098
Total energy intake, kcal/day	1736 (591)	1421 (374)	1375 (352)	2067 (573) * †	2089 (589) * †	<0.001
Energy intake, kcal/IBW/day	30.7 (9.8)	25.1 (5.3)	23.8 (5.2)	37.2 (9.0) * †	37.2 (9.4) * †	<0.001
Total protein intake, g/day	74.7 (29.9)	53.6 (11.2)	53.0 (11.3)	95.8 (28.6) * †	96.8 (26.6) * †	<0.001
Animal protein intake, g/day	46.5 (23.5)	31.4 (9.5)	29.4 (10.4)	61.8 (22.2) * †	63.8 (21.8) * †	<0.001
Animal protein intake, g/IBW/day	6.02 (3.05)	4.05 (1.20)	3.76 (1.30)	8.04 (2.79) * †	8.27 (2.85) * †	<0.001
Vegetable protein intake, g/day	28.2 (9.8)	22.3 (7.1)	23.6 (6.0)	34.1 (10.8) * †	33.1 (9.0) * †	<0.001
Vegetable protein intake, g/IBW/day	3.63 (1.21)	2.86 (0.84)	3.01 (0.73)	4.43 (1.31) * †	4.27 (1.08) * †	<0.001
Protein intake, g/IBW/day	1.33 (0.54)	0.95 (0.17)	0.92 (0.19)	1.73 (0.49) * †	1.73 (0.48) * †	<0.001
Adequate protein intake (−/+)	108/106	45/0	46/0	0/34	0/61	<0.001
Total fat intake, g/day	55.1 (21.9)	43.3 (12.6)	40.6 (11.2)	65.2 (17.6) * †	70.5 (23.7) * †	<0.001
Fat intake, g/IBW/day	0.98 (0.39)	0.77 (0.20)	0.70 (0.18)	1.18 (0.31) * †	1.26 (0.42) * †	<0.001
Total carbohydrate intake, g/day	218.2 (81.7)	191.1 (67.2)	182.8 (59.4)	251.5 (97.6) * †	249.4 (79.3) * †	<0.001
Carbohydrate intake, g/IBW/day	3.9 (1.3)	3.4 (1.1)	3.2 (1.0)	4.5 (1.6) * †	4.4 (1.3) * †	<0.001
Appendicular muscle mass, kg	17.6 (3.8)	17.4 (3.8)	18.1 (3.6)	17.2 (3.7)	17.8 (4.1)	0.673
Body fat mass, kg	17.6 (7.4)	19.0 (7.7)	17.3 (7.8)	17.6 (7.5)	16.7 (6.6)	0.431
Percent body fat mass, %	28.4 (8.8)	30.3 (9.0)	27.8 (8.6)	28.7 (9.0)	27.4 (8.5)	0.313
SMI, kg/m^2^	6.8 (0.9)	6.8 (0.9)	6.9 (0.9)	6.8 (0.8)	6.9 (1.1)	0.767
Change in SMI, kg/m^2^/month	−0.007 (0.023)	−0.007 (0.020)	−0.013 (0.019)	−0.011 (0.027)	0.001 (0.023) † ‡	0.002
Rate of SMI change, %	−1.14 (4.10)	−1.22 (3.71)	−2.31 (3.30)	−1.88 (4.62)	0.36 (4.29) † ‡	0.002
Decreasing SMI (−/+)	87/127	19/33	15/41	14/26	39/27	0.002

Adequate protein intake was defined as protein intake ≥1.2 g/kg IBW/day. Data are expressed as mean (standard deviation) or counts. The differences between groups were evaluated using Student’s t-test or chi-square test. eGFR, estimated glomerular filtration rate; IBW, ideal body weight; SMI, skeletal muscle mass index. Change in SMI (kg/m^2^/month) was calculated as (SMI at follow-up examination (kg/m^2^) minus SMI at baseline examination (kg/m^2^)) divided the follow up durations (month). Rate of SMI change (%) was calculated as change of SMI (kg/m^2^/month) × 12/SMI at baseline examination (kg/m^2^). Decreasing SMI was defined as a rate of SMI change (%)under −0.5%. The difference between groups was evaluated with one-way ANOVA and Tukey-Kramer test, or chi-square test. * *p* < 0.05 vs. Exercise (−)/Adequate protein intake (−); † *p* < 0.05 vs. Exercise (+)/Adequate protein intake (−); and ‡ *p* < 0.05 vs. Exercise (−)/Adequate protein intake (+).

**Table 2 nutrients-12-03220-t002:** Odds ratio for decreasing SMI.

	Model 1		Mode 2		Model 3	
	OR (95% CI)	*p*	OR (95% CI)	*p*	OR (95% CI)	*p*
Men	1.22 (0.66–2.25)	0.520	0.40 (0.15–1.09)	0.072	0.43 (0.16–1.18)	0.095
Age, years	0.98 (0.92–1.03)	0.386	0.99 (0.93–1.05)	0.739	0.99 (0.93–1.05)	0.740
Smoking, yes	0.76 (0.32–1.79)	0.533	0.75 (0.31–1.80)	0.519	0.72 (0.30–1.76)	0.476
Alcohol consumption, yes	1.04 (0.36–3.05)	0.930	1.15 (0.38–3.47)	0.805	2.38 (0.25–22.6)	0.449
Duration of diabetes, years	―	―	1.01 (0.97–1.04)	0.730	1.01 (0.98–1.04)	0.695
SMI at baseline examination, kg/m^2^	―	―	2.49 (1.37–4.55)	0.002	2.40 (1.29–4.47)	0.004
BMI, kg/m^2^	―	―	0.92 (0.83–1.03)	0.159	0.93 (0.82–1.04)	0.194
Energy intake, kcal/IBW/day	―	―	0.98 (0.94–1.02)	0.342	0.86 (0.62–1.20)	0.388
Animal proteins intake, kcal/IBW/day	―	―	―	―	1.17 (0.87–1.56)	0.302
Vegetable proteins intake, kcal/IBW/day	―	―	―	―	1.03 (0.61–1.74)	0.915
Carbohydrate intake, kcal/IBW/day	―	―	―	―	1.85 (0.42–8.21)	0.419
Fat intake, kcal/IBW/day	―	―	―	―	1.62 (0.06–41.9)	0.773
Exercise (−)/Adequate protein intake (−)	2.58 (1.21–5.48)	0.014	1.21 (0.22–6.74)	0.829	2.50 (0.90–6.90)	0.078
Exercise (+)/Adequate protein intake (−)	3.77 (1.73–8.19)	<0.001	3.37 (1.28–8.85)	0.014	3.58 (1.24–10.4)	0.019
Exercise (−)/Adequate protein intake (+)	2.70 (1.19–6.15)	0.018	1.60 (0.73–3.51)	0.245	3.03 (1.27–7.22)	0.012
Exercise (+)/Adequate protein intake (+)	Ref	―	Ref	―	Ref	―

SMI, skeletal muscle mass index; OR, odds ratio; CI, confidence interval. Decreasing SMI was defined as a rate of SMI change (%) under −0.5%.

**Table 3 nutrients-12-03220-t003:** Multiple regression analysis on the rate of SMI change according to protein intake.

Adequate Protein Intake (−) *n* = 108	Standardized β	*p*
Men	0.06	0.679
Age, years	−0.001	0.990
Duration of diabetes, years	−0.010	0.929
Smoking, yes	−0.145	0.201
Alcohol consumption, yes	−0.005	0.978
SMI at baseline examination, kg/m^2^	−0.014	0.940
Body mass index, kg/m^2^	−0.207	0.205
Energy intake, kcal/IBW/day	0.236	0.672
Animal protein intake, kcal/IBW/day	−0.129	0.435
Vegetable protein intake, kcal/IBW/day	−0.124	0.547
Carbohydrate intake, kcal/IBW/day	−0.102	0.835
Fat intake, kcal/IBW/day	0.036	0.860
Exercise habit, yes	−0.182	0.094
**Adequate protein intake (+) *n* = 106**	**Standardized β**	***p***
Men	0.298	0.073
Age, years	0.039	0.704
Duration of diabetes, years	−0.021	0.845
Smoking, yes	0.134	0.189
Alcohol consumption, yes	−0.011	0.969
SMI at baseline examination, kg/m^2^	−0.427	0.022
Body mass index, kg/m^2^	0.011	0.969
Energy intake, kcal/IBW/day	0.207	0.862
Animal protein intake, kcal/IBW/day	0.022	0.934
Vegetable protein intake, kcal/IBW/day	0.028	0.871
Carbohydrate intake, kcal/IBW/day	−0.140	0.860
Fat intake, kcal/IBW/day	−0.028	0.953
Exercise habit, yes	0.255	0.011

Adequate protein intake was defined as protein intake ≥1.2 g/kg IBW/day. IBW, ideal body weight; SMI, skeletal muscle mass index. Rate of SMI change (%) was calculated as change of SMI (kg/m^2^/month) × 12/SMI at baseline examination (kg/m^2^). Gender was defined as women (=0) or men (=1), smoking status was defined as non-smoker (=0) or smoker (=1), alcohol consumption was defined as non-consumer (=0) or consumer (=1), and exercise habit was defined as no exercise habit (=0) or exercise habit (=1).

## Data Availability

The datasets used and/or analyzed during the current study are available from the corresponding author on reasonable request.

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
