# Peer review of "Effect of Exercise Habit on Skeletal Muscle Mass Varies with Protein Intake in Elderly Patients with Type 2 Diabetes: A Retrospective Cohort Study"

_nutrients, 2020, doi:10.3390/nu12103220_

Round 1
Reviewer 1 Report
The present manuscript provides evidence from a retrospective cohort study supporting the conclusion that regular exercise in the elderly without adequate dietary protein intake can actually intensify the loss of muscle mass typically observed in aging.
This main message of the article is valuable and has merit. The authors are careful to note that the limitations of the study include the retrospective nature of the design and lack of information about the exercise protocols undertaken by the subjects. An additional limitation not mentioned concerns the inaccuracy associated with dietary questionnaires involving recall of the subjects. In a retrospective design the authors could not control for this potential confounding factor but it should nonetheless be mentioned.
The lack of detailed information about the frequency, intensity, duration, and type of exercise performed by the subjects also limits the strength of any interpretations we can make regarding the data. Resistance versus aerobic exercise has profoundly different effects on preservation and increase of muscle mass, which is the main question this study seeks to address. There is little detailed information regarding the nature of exercise performed by the subjects and therefore a great deal of caution is warranted when interpreting these results.
Author Response
Response to Reviewer 1 Comments
The present manuscript provides evidence from a retrospective cohort study supporting the conclusion that regular exercise in the elderly without adequate dietary protein intake can actually intensify the loss of muscle mass typically observed in aging.
Thank you for your comments. As you kindly suggested, our manuscript was revised as follows.
Point 1. This main message of the article is valuable and has merit. The authors are careful to note that the limitations of the study include the retrospective nature of the design and lack of information about the exercise protocols undertaken by the subjects. An additional limitation not mentioned concerns the inaccuracy associated with dietary questionnaires involving recall of the subjects. In a retrospective design the authors could not control for this potential confounding factor but it should nonetheless be mentioned.
Response
Thank you for your constructive suggestion. As you say, the inaccuracy associated with dietary questionnaires involving recall of the subjects and a retrospective design are limitations of this study. According to your suggestion, we have added these points as the limitations of this study descried as below.
“First, the study design is a retrospective cohort study. Therefore, there may be unknown confounders. Second, the inaccuracy associated with dietary questionnaires involving recall of the subjects, although the BDHQ showed reasonable validity with semi-weighed dietary records [24].”
- Kobayashi S, Murakami K, Sasaki S, et al. Comparison of relative validity of food group intakes estimated by comprehensive and brief-type self-administered diet history questionnaires against 16 d dietary records in Japanese adults. Public Health Nutr. 2011;14:1200-1211.
Point 2. The lack of detailed information about the frequency, intensity, duration, and type of exercise performed by the subjects also limits the strength of any interpretations we can make regarding the data. Resistance versus aerobic exercise has profoundly different effects on preservation and increase of muscle mass, which is the main question this study seeks to address. There is little detailed information regarding the nature of exercise performed by the subjects and therefore a great deal of caution is warranted when interpreting these results.
Response
Thank you for your constructive comment. As you say, the lack of detailed information about the frequency, intensity, duration, and type of exercise performed by the subjects is also a limitation of this study, because the frequency, intensity, duration, and type of exercise has profoundly different effects on preservation and increase of muscle mass. According to your comment, we have added this point in the Discussion section described as below.
“Third, the lack of detailed information about the frequency, intensity, duration, and type of exercise performed by the subjects is also a limitation of this study, because the frequency, intensity, duration, and type of exercise has profoundly different effects on preservation and increase of muscle mass.”
Reviewer 2 Report
MAJOR COMMENTS:
In the manuscript entitled “Effect of exercise habits on skeletal muscle mass varies with protein intake: A retrospective cohort study”, Hashimoto et al. provided interesting findings that dietary protein supplement is required for elderly diabetic patients to augment exercise-induced muscle mass. Authors missed several important information that need to be presented. Furthermore, the current information does not support authors’ conclusion.
- Authors addressed very important question whether appropriate protein intake is important to augment the exercise effect to increase muscle mass in aged type 2 diabetic patients. Although the scientific question is very important and authors presented some interesting results, this reviewer was not convinced that the current study would advance our current knowledge. Most importantly, the exercise assessment should have been done so we can completely exclude the possibility that the difference in SMI change in exercise groups with or without protein intake is due to the protein supplement.
- Although authors proposed several possibility and explanation, the mechanical studies need to be done to get more advanced knowledge.
- As authors pointed out, the lack of protein intake information leads this study to be inconclusive.
MIMOR COMMENTS:
- Abstract was hard to follow and somewhat confusing. Reorganizing the results will be helpful to present the scientific conclusion better.
- 73 exclusion was explained in the Fig. 1. However, there were only 53 people listed.
- There were several grammatical errors that need to be edited.
Author Response
Response to Reviewer 2 Comments
In the manuscript entitled “Effect of exercise habits on skeletal muscle mass varies with protein intake: A retrospective cohort study”, Hashimoto et al. provided interesting findings that dietary protein supplement is required for elderly diabetic patients to augment exercise-induced muscle mass. Authors missed several important information that need to be presented. Furthermore, the current information does not support authors’ conclusion.
Thank you for your comments. As you kindly suggested, our manuscript was revised as follows.
MAJOR COMMENTS:
Point 1. Authors addressed very important question whether appropriate protein intake is important to augment the exercise effect to increase muscle mass in aged type 2 diabetic patients. Although the scientific question is very important and authors presented some interesting results, this reviewer was not convinced that the current study would advance our current knowledge. Most importantly, the exercise assessment should have been done so we can completely exclude the possibility that the difference in SMI change in exercise groups with or without protein intake is due to the protein supplement.
Response
Thank you for your valuable comment. As you say, the detail exercise assessment was important. Unfortunately, however, we did not have a detail data of exercise as we mentioned in the Limitation. According to your comment, we have revised the Limitation described as below.
“Third, the lack of detailed information about the frequency, intensity, duration, and type of exercise performed by the subjects is also a limitation of this study, because the frequency, intensity, duration, and type of exercise has profoundly different effects on preservation and increase of muscle mass.”
Point 2. Although authors proposed several possibility and explanation, the mechanical studies need to be done to get more advanced knowledge.
Response
Thank you for your valuable comment. As you say, the mechanical studies need to be done to get more advanced knowledge. According to your comment, we have added this point in the Discussion section described as below.
“However, further studies needed to be done to get more advanced knowledge for the association among protein intake, exercise and muscle mass.”
Point 3. As authors pointed out, the lack of protein intake information leads this study to be inconclusive.
Response
Thank you for your comment. As you say, the lack of protein intake information, such as the timing and the quality of protein intake, is also important information, as we mentioned in the Limitation. However, we showed that the presence or absence of adequate daily protein intake had an effect on changes in muscle mass. According to your comment, we have added this point in the limitation described as below.
“Fourth, the timing and the quality of protein intake, which were also important factor for the sarcopenia [31], was not evaluated. However, we showed that the presence or absence of adequate daily protein intake had an effect on changes in muscle mass.”
- Coelho-Junior HJ, Marzetti E, Picca A, Cesari M, Uchida MC, Calvani R. Protein Intake and Frailty: A Matter of Quantity, Quality, and Timing. Nutrients. 2020;12:E2915.
MIMOR COMMENTS:
Point 1. Abstract was hard to follow and somewhat confusing. Reorganizing the results will be helpful to present the scientific conclusion better.
Response
Thank you for your constructive comment. According to your comment, we have revised the Abstract.
Point 2. 73 exclusion was explained in the Fig. 1. However, there were only 53 people listed.
Response
Thank you for your comment and we are sorry for confusing you. In this study, 53 people were excluded. We have revised the Figure 1 and the Result section described as below.
“Fifty-three patients were excluded; and thus, finally study participants consisted of 214 participants (Figure 1).”
Point 3. There were several grammatical errors that need to be edited.
Response
Thank you for your constructive comment. According to your comment, we have checked and revised the manuscripts.
Reviewer 3 Report
This is a retrospective, observational and descriptive study. The study design is longitudinal but retrospective.The observation period is 18 months only. The study population is relatively small, it is about patients with type 2 Dm which is not mentioned in the title of the ms.The authors try to oder the decreases in SMI by a no of variables including self-reported exercise and diet (including protein intake). The major conclusion refers to a combination of exercise and sufficient protein intake to prevent loss in SM.
Data presentation looks a little bit chaotic. The precision of the methods used (e.g., to assess SM and protein intake) is not given and has to be taken into account for a careful interpretation of the data. In addition, the validation of the BIA device to assess SM has to be documented.
The study protocol is uncontrolled, all informations depend on self-reports (which have a well known bias). Thus, these informations cannot serve as a valid basis for calculation.
It looks like that SMI decreases during the observation period. This is independent of the model used. The major determinant of the decrease of SM is the basal SM suggesting that during the observation period exercise and protein intake are of minor importance. This is contrary to the position of the authors.
Since this an observational study the authors cannot do any conclusions regarding suitable interventions.
Author Response
Response to Reviewer 3 Comments
Thank you for your comments. As you kindly suggested, our manuscript was revised as follows.
Point 1. This is a retrospective, observational and descriptive study. The study design is longitudinal but retrospective. The observation period is 18 months only. The study population is relatively small, it is about patients with type 2 Dm which is not mentioned in the title of the ms. The authors try to oder the decreases in SMI by a no of variables including self-reported exercise and diet (including protein intake). The major conclusion refers to a combination of exercise and sufficient protein intake to prevent loss in SM.
Response
Thank you for your valuable comments. According to your comments, we have mentioned type 2 diabetes in the Title described as below.
“Effect of exercise habits on skeletal muscle mass varies with protein intake in elderly patients with type 2 diabetes: A retrospective cohort study”
In addition, as you say, follow-up duration of 18 months was not so long and the number of participants were not so large; thus, we have mentioned this point as one of the limitations of this study described as below.
“In addition, mean study duration was 18 months and this follow-up duration might not be enough.”
“Lastly, the number of participants was not so large; thus, further studies with larger number of participants are needed.”
Point 2. Data presentation looks a little bit chaotic. The precision of the methods used (e.g., to assess SM and protein intake) is not given and has to be taken into account for a careful interpretation of the data. In addition, the validation of the BIA device to assess SM has to be documented.
Response
Thank you for your valuable suggestion. As you say, the precision of the methods used is an important information. Median Spearman's correlation coefficients of BDHQ and the semi-weighed dietary records were reported to be 0.48 for men and 0.44 for women. In addition, the R2 coefficients of BIA and DEX were reported to be 0.88 for men and 0.83 for women. According to your suggestion, we have added these points in the Methods section described as below.
“The R2 coefficients of BIA and DEX were 0.88 for men and 0.83 for women [20].”
“The median Spearman's correlation coefficients of BDHQ and semi-weighed dietary records were 0.48 for men and 0.44 for women [24].”
- Kobayashi S, Murakami K, Sasaki S, et al. Comparison of relative validity of food group intakes estimated by comprehensive and brief-type self-administered diet history questionnaires against 16 d dietary records in Japanese adults. Public Health Nutr. 2011;14:1200-1211.
Point 3. The study protocol is uncontrolled, all informations depend on self-reports (which have a well known bias). Thus, these informations cannot serve as a valid basis for calculation.
Response
Thank you for your comment. As you say, the data of BDHQ and exercise habit depended on self-reports. According to your comment, we have added these points as the limitations of this study in the Discussion section described as below.
“Second, the inaccuracy associated with dietary questionnaires involving recall of the subjects, although the BDHQ showed reasonable validity with semi-weighed dietary records [24]. Third, the lack of detailed information about the frequency, intensity, duration, and type of exercise performed by the subjects is also a limitation of this study, because the frequency, intensity, duration, and type of exercise performed by the subjects is also a limitation of this study, because the frequency, intensity, duration, and type of exercise has profoundly different effects on preservation and increase of muscle mass.”
Point 4. It looks like that SMI decreases during the observation period. This is independent of the model used. The major determinant of the decrease of SM is the basal SM suggesting that during the observation period exercise and protein intake are of minor importance. This is contrary to the position of the authors.
Response
Thank you for your comment. As you say, SMI decreasesd during the observation period, the rate of SMI change (%) was 1.14%. Previous study showed that muscle mass decreasesd with age, especially in the elderly, by 0.5% to 1.0% per year. In this study, we showed that the rate of SMI change in non- habitual exercisers with inadequate protein intake (-1.22% [SD 3.71]) was almost the same as natural history of decrease of SMI and that in habitual exercisers with inadequate protein intake (-2.26% [SD 3.59]) and non-habitual exercisers with adequate protein intake (-1.88% [SD 4.62]) were significantly worse than that in habitual exercisers with adequate protein intake (0.36% [SD 4.29]). Furthermore, to minimize the effect of basal SMI, multiple regression analysis on the change in SMI (kg/m2/year) according to the amount of protein intake was also performed. The results are almost the same as an original manuscript.
Multiple regression analysis on the change in SMI (kg/m2/year) according to the amount of protein intake
|
Adequate protein intake (-) n = 108 |
Standardized β |
p |
|
Men |
0.067 |
0.652 |
|
Age, years |
0.018 |
0.865 |
|
Duration of diabetes, years |
-0.019 |
0.853 |
|
Smoking, yes |
-0.140 |
0.186 |
|
Alcohol consumption, yes |
0.045 |
0.670 |
|
SMI at baseline examination, kg/m2 |
-0.047 |
0.783 |
|
Body mass index, kg/m2 |
-0.219 |
0.117 |
|
Exercise habit, yes |
-0.197 |
0.0496 |
|
Adequate protein intake (+) n = 106 |
Standardized β |
P |
|
Men |
0.366 |
0.024 |
|
Age, years |
0.004 |
0.972 |
|
Duration of diabetes, years |
-0.020 |
0.840 |
|
Smoking, yes |
0.223 |
0.119 |
|
Alcohol consumption, yes |
0.038 |
0.707 |
|
SMI at baseline examination, kg/m2 |
-0.516 |
0.004 |
|
Body mass index, kg/m2 |
0.111 |
0.403 |
|
Exercise habit, yes |
0.261 |
0.006 |
Point 5. Since this an observational study the authors cannot do any conclusions regarding suitable interventions.
Response
Thank you for your valuable suggestion. As you say, since this is an observational study, our conclusions were not suitable for this study. According to your suggestion, we have revised the conclusion described as below.
“Therefore, elderly patients with habitual exercise habits without adequate protein intake should be aware of the loss of muscle mass.”
Round 2
Reviewer 1 Report
The authors have addressed the comments successfully.
Author Response
Thank you very much for providing important comments. We are thankful for the time and energy you expended.
Reviewer 2 Report
Although authors intensively responded to the reviewers' critique, the scientific integrity of the manuscript has not been improved. Authors only revised manuscript in a limited scope.
Author Response
Response to Reviewer 2
Although authors intensively responded to the reviewers' critique, the scientific integrity of the manuscript has not been improved. Authors only revised manuscript in a limited scope.
Response
We thank you very much for your advices to improve our manuscript. As you say, there still remains the problems and limitations. However, our data showed that exercise habits without adequate protein intake were associated with the increases risk of decreasing of SMI compared to exercise habits with adequate protein intake, which we would inform people involved in medical care.
Reviewer 3 Report
This is ok now.
Author Response

(The authors gave the same response as above.)
